# Effects of a land and aquatic exercise-based program on pain, mobility and quality of life in patients with chronic low back pain: a study protocol for a randomized controlled trial

**Joana Borges**[1,2]* , **Diogo Monteiro**[1,3] , **Fernanda M. Silva**[4,5] , **Miguel Jacinto**[1,3] , **Tiago Pastilha**[6] , **Pedro Duarte-Mendes**[7,8]*

**1** ESECS - Polytechnic University of Leiria, Leiria, Portugal, **2** Department of Physical Medicine and Rehabilitation, Unidade Local de Saúde do Baixo Mondego, Figueira da Foz, Portugal, **3** Research Center in Sport Sciences, Health Sciences and Human Development (CIDESD), Covilhã, Portugal, **4** University of Coimbra, CIPER, Faculty of Sport Sciences and Physical Education, Coimbra, Portugal, **5** School of Education and Communication, University of Algarve, Faro, Portugal, **6** Department of Pulmonology, Unidade Local de Saúde do Baixo Mondego, Figueira da Foz, Portugal, **7** Department of Sports and Well-Being, Polytechnic Institute of Castelo Branco, Castelo Branco, Portugal, **8** Sport Physical Activity and Health Research & INnovation CenTer (SPRINT), Santarém, Portugal

☢ These authors contributed equally to this work.
* joanaccborges24@hotmail.com (JB); pedromendes@ipcb.pt (PD-M)

**Data availability statement:** No datasets were generated or analysed during the current study.

## Abstract

### Background

Chronic low back pain (CLBP) is a disease with significant functional, emotional and social impact. Several interventions are proposed for its management and exercise is one of main, land-based or water-based. This study describes a randomized controlled trial that will analyze the effect of a combined aquatic and land-based exercise program compared to an aquatic-based program on pain, functional incapacity and quality of life in adults with CLBP. Additionally, it will analyze the effects of exercise cessation in the same outcomes.

### Methods and design

A blind randomized controlled trial will be developed with a 1:1 allocation ratio. Around 30 adults with mechanical CLBP will be randomly allocated in two groups. The experimental group (ALG) will complete an aquatic and land-based exercise program and control group (AG) will carry out only an aquatic program, both for 8 weeks. Participants will be assessed with Visual Analogue Scale, Oswestry Disability Index, Short-Form 36, Tampa Scale of Kinesiophobia-13 and Modified-Modified Schober Test, collected at baseline (M0), after 8 weeks (M1) and 4 weeks after the end of the intervention (M2).

All relevant data from this study will be made available upon study completion.

**Funding:** This work was funded by National Funds by FCT - Foundation for Science and Technology under the following project UID/04045: Research Center in Sports Sciences, Health Sciences, and Human Development. The funders had no role in study design, data collection and analysis, decision to publish, or preparation of the manuscript.

**Competing interests:** The authors have declared that no competing interests exist.

## Discussion

This study may provide a relevant contribution to understand the potential effect of a combined land and aquatic exercise program on pain, functional disability, fear of movement, quality of life and lumbar mobility. The results may provide important information for CLBP management.

### Trial registration

This trial is registered with ClinicalTrials.gov (registration number: NCT06641570; date of registration: October 14, 2024).

## Introduction

Chronic low back pain (CLBP) is considered as "pain tension or stiffness localized below the costal margin and above the inferior gluteal folds", lasting more than 12 weeks [1–3]. It can be distinguished in two main categories: specific, when the cause is attributed to a pathology or structural lesion and non-specific (accounts for about 90% of cases), when it is not possible to determine a recognizable known cause [3]. Low physical activity levels, obesity, smoking and high stress constitute risk factors for its occurrence [4].

This is one of the most frequent public health problems with a high prevalence in the world population [5]. According to GBD 2021 [3], CLBP in 2020 affected around 619 million people worldwide, and it is estimated that more than 80% of the population will have at least one episode of low back pain in their life [5]. The prevalence of CLBP is 4.2% in people aged between 24 and 39 years old and 19.6% between 20 and 59 years old, noting that women have CLBP more frequently than men [6]. Therefore, CLBP has a strong impact on functionality, well-being and quality of life, leading to limitations in social and work life of who suffers from this health condition, with economy implications and direct and indirect annual costs associated with health care resources usage [7–9].

These justify the need to study therapeutic strategies to manage this chronic condition. Some guidelines for the management of CLBP recommend physical exercise as a first-line treatment to reduce pain and disability, with programs duration superior than twenty hours for higher benefits, not mentioning the most appropriate type or intensity [10–12]. A systematic review with meta-analysis conducted by Hayden et al. [13] concluded that exercise can provide benefits in pain and disability caused by CLBP, compared to controls without treatment or with placebo intervention. Additionally, the same authors added that few studies report side effects associated with exercise.

On the other hand, prolonged periods of physical inactivity negatively impact the recovery from pain and functional disability caused by CLBP, so it is recommended that these individuals remain physically active [14]. Some authors suggest a variety of exercise types for managing CLBP, including aerobic exercise (from light to high intensity) [15,16], muscle strengthening exercises [17,18], and flexibility exercises [19]. Furthermore, Psycharakis et al. [20] highlight core stability exercises, as they recruit lumbar spine and pelvis stabilizing muscles. These exercises aim to counteract the decreased spine and pelvis neuromotor control and the muscle weakness of the abdominal and hip muscles observed in individuals with CLBP [21,22]. However, there is no consensus on the most suitable type of exercise, making it a controversial topic that requires further scientific research [23,24].

The reduction of lumbar spine stability due to decreased core muscle activity can lead to CLBP, with some studies indicating that activation exercises for the abdominal and spinal extensor muscles may play an important role in this condition management [25–27]. In turn, decreased lumbar spinal joint mobility and reduced flexibility in muscles such as the spinal extensors, hip flexors, and hamstrings may also contribute to CLBP persistence, justifying the inclusion of stretching exercises in exercise programs [26,28].

Another proposed modality is aquatic exercise, which utilizes the properties and characteristics of water for physical activity, aiming to reduce pain, strengthen muscles, and promote relaxation, among other benefits [29,30]. Thus, aquatic exercise is of particular interest, as the unique properties of water, such as buoyancy and hydrostatic pressure, reduce joint stress on the spine and other joints, and enhance balance, mobility, and pain control [29,31,32]. Additionally, the aquatic environment allows individuals to perform movements that are typically difficult or impossible on land [33]. Beyond this, continuous movement of the limbs against the resistance of the water leads to increased muscle strength and cardiovascular benefits [34–36]. A systematic review by Ma et al. [37] found that aquatic exercise programs were effective in reducing pain and improving quality of life and functional capacity in patients with CLBP, compared to other interventions or control groups with no intervention. Some authors argue that the buoyancy achieved during aquatic exercise can reduce the impact of gravity on the body, decreasing joint stress during exercise and pain [38–40]. Other researchers found that aquatic may be more beneficial than land-based programs in lumbar pain, health-related quality of life, and functional disability [41–43]. In contrast, Yalfani et al. [44] compared a Pilates exercise program performed in water with the same program conducted on land and found that both significantly reduced pain and dysfunction, with no significant differences between the programs. In light of these differences in the literature, no study has yet investigated whether there is an additional effect from combining land-based and water-based exercises on the same variables, compared to an exclusive water exercise program.

Moreover, few authors have investigated the effects of cessation of exercise [45,46]. Mujika & Padilla [46] argue that detraining can be divided into two study periods: short-term, less than 4 weeks, and long-term, more than 4 weeks. Therefore, Toraman & Ayceman [47] found that after 4 weeks of interruption following a 9-week aerobic, strength, and flexibility training program, conducted 3 times a week, participants showed a reduction in performance levels on various functional tests, standing out the Chair Sit and Reach, compared to the end of the intervention, although the changes were not statistically significant. Thus, the impact of aquatic exercise, core stability exercise, and trunk and lower limb stretching cessation on individuals with CLBP remains unclear.

This trial aims to assess the effect of a combined aquatic and land-based exercise program compared to an aquatic program on pain, functional disability, and quality of life in adults with CLBP. Secondary objectives include investigating the effects on lumbar mobility and fear of movement, as well as the impact of training cessation on these variables 4 weeks after the end of the intervention. We hypothesize that combined aquatic and land-based exercise decreases significantly pain, and functional disability, and increases quality of life in adults with CLBP.

## Materials and methods

### Study design

This investigation will follow a quantitative model and a randomized controlled trial (RCT) study design, according to the Statement guidelines for standard protocol items in clinical

trials (SPIRIT) and the Consolidated Standards of Reporting Trials (CONSORT) statement [48,49]. S1 Fig shows the SIPRIT checklist. This study protocol is registered in the Clinicaltrials.gov ID: NCT06641570 (date of registration: October 14, 2024).

## Participants

The study sample will consist of both gender Portuguese participants, over 18 years, with CLBP lasting three months or longer and with an intensity of 3 or higher on the visual analogue scale (VAS) at rest. Individuals who refuse to participate or do not provide informed consent will be excluded, as well as those with pain radiating to the lower limb, recent pregnancy or childbirth (less than 8 weeks ago) [50], ongoing physical therapy treatments, severe rheumatological, neurological, neoplastic, cardiovascular diseases, or other conditions that could prevent full participation in the intervention, a history of spinal surgery, inflammatory, infectious, or malignant spinal diseases, and psychiatric disorders that may affect adherence and symptom assessment [41–43].

The sample will be randomly divided into two groups: the experimental group, which will undergo the aquatic and land-exercise program (ALG), and the control group, which will perform the aquatic-exercise program (AG).

The required sample size was calculated using the G*Power software (3.1.9.4; Heinrich Heine University Düsseldorf, DE), based on a study with 65 participants by Dundar et al. [42] and corroborated by other studies [51–53]. The aim was to detect differences in pain intensity (VAS) between the aquatic exercise program group and the control group (land exercise), with an effect size of 1.079, $\alpha$=0.05, and a power of 80%. The total sample size needed to achieve the objective will be 30 participants (15 in each group).

Thus, the study should start with an additional 15% of the planned sample size for each group to account for potential dropouts and as well as adopt cognitive-behavioral strategies to reduce dropout rates, such as setting flexible and participant-driven goals or providing positive and individualized feedback [54,55].

## Ethics

According to the Helsinki Declaration protocol (1964) [56], written informed consent will be obtained from the participants. All participants receive a signed copy. Each participant will receive a unique coded alphanumeric identification ensuring data anonymity in accordance with the General Data Protection Regulation (GDPR), following Regulation (EU) 2016/679 of the European Parliament of April 27, 2016, and the Personal Data Protection Law, according to Law No. 58/2019 of August 8. The files that connect alphanumeric codes to participants identifying information will be stored in a separate locked file protected by a password access system, only known by the principal investigator [48].

This study will be based on the ethical standards required in scientific research and was submitted for review by the Ethics Committee of the Polytechnic University of Leiria and approved on March 27, 2024 (reference: CE/IPLEIRIA/47/2024). Study procedures, protocol risks voluntary character of participation and expected results will be explained to the participants. The right to withdraw without any prejudice or consequence will be ensured.

## Intervention

The intervention will take place over 8 weeks at the Municipal Pool of Paião, located in Figueira da Foz, Portugal. The AG will conduct the aquatic exercise program, while the ALG will carry out both aquatic and land-based exercise programs. The intervention will begin

after a one-week washout period, during which participants will cease any additional physical exercise, which they will maintain throughout the intervention weeks.

To monitor the intensity of the interventions, Borg CR-10 scale will be used. This is a tool to rate dyspnea and/or perceived exertion (RPE) experienced by an individual. The scale consists of a numerical list with 10 points (0–10), an adaptation of the original scale (6–20), that assesses and monitors the intensity/severity of shortness of breath/fatigue. Its use requires explaining that "0" means not experiencing dyspnea/fatigue at the moment and that "10" means the worst dyspnea/fatigue they have ever experienced or the worst imaginable dyspnea/fatigue [57,58]. This instrument will not be used during the assessment moments before and after the exercise programs but only to monitor the intensity during the sessions of both aquatic and land-based exercise programs. This scale is validated for monitoring dyspnea and fatigue, showing an excellent correlation with heart rate ($r = 0.92$) and VO2 ($r=0.92–0.93$) and moderate reliability (ICC=0.50–0.66) [59].

Participants will be encouraged to maintain their daily activities as usual outside of the study. The participants may not participate in any other exercise programs.

**Aquatic-exercise program.** This intervention will consist of an aquatic-exercise program (Table 1), administered by a physiotherapist. It will include 16 sessions, twice a week, over 8 weeks at a water temperature between 32–34°C. Each session will be conducted in groups of 8–12 patients and will last about 45 minutes. The aquatic exercise sessions will include a warm-up phase (5 minutes): walking forwards, sideways, and backwards; a training phase (35 minutes): active range of motion exercises (10 minutes) for the spine, upper and lower limbs; aerobic exercises (5 minutes) with jumps and static running at various speeds; muscle strengthening (10–15 minutes), (8–12 repetitions), 1–2 sets, 8–10 exercises including hip flexion, extension, adduction, and abduction, knee flexion and extension, knee cycling, shoulder flexion, extension, and abduction, elbow flexion, extension, and pronation-supination, squatting, and stretching exercises (5–10 minutes) for the neck, trunk, and limbs; and a relaxation/cool-down phase (5 minutes): including lying supine and low-impact exercises such as slow walking, squatting, and standing [20,37,38,41,42]. For participants who cannot complete any exercise due to difficulty or discomfort, a modified version of each exercise will be offered. The load progression will be managed using auxiliary equipment, including flotation devices, plates, balls, pull-buoys, and dumbbells, according to each participant's RPE on the Borg CR-10 scale.

**Land-exercise program.** This program will include specific land-based exercises (Table 2, guided by a physiotherapist. This intervention will include 8 sessions in 8 weeks, once a week, on a different day from the aquatic exercise program. Each session will last between 25 to 35 minutes and will be divided into two components: core stability exercises and stretching of the trunk and lower limbs [60]. Core stability exercises (15–20 minutes): these will consist of isometric exercises which contraction was maintained for 7–8 seconds, 8–12 repetitions, 1 set, with brief rest intervals of 3 seconds between repetitions and 30 seconds to 1 minute between exercises [60,61]. Exercises will include abdominal hollowing, side bridge (both sides), supine extension bridge, lower limb extension from prone (both sides), and alternate arm and leg raise from quadruped. Intensity will increase gradually by reducing rest time between exercises and increasing repetitions, based on participants' performance. Participants will be instructed to contract and hold their abdominal muscles while maintaining normal breathing patterns [62]. Stretching exercises (10–15 minutes): participants will perform muscle stretching exercises, holding the maximum stretch for 30 seconds to 1 minute, and returning to the original position, followed by 5–10 seconds of rest. Each exercise will be repeated two to three times [63]. Stretches will target the hamstrings, iliopsoas, piriformis, and tensor fasciae latae.

**Table 1. Aquatic-exercise program.**

| Type | Frequency and duration | Intensity | |
|---|---|---|---|
| | | Week 1–4 | Week 5–8 |
| **Aquatic exercise:** aerobic, resistance, and flexibility training performed in water. | 2 times per week. Session duration: 40–45 min (warm-up phase: 5 min, training phase: 30–35 min; and relaxation/cool-down phase: 5 min). | **Warm-up phase:** Low-intensity aerobic exercise: walking forwards, sideways, and backwards. RPE: 3–4 (Borg CR-10 Scale). **Training phase:** Active range of motion exercises: spine, upper and lower limbs. RPE: 3–4 (Borg CR-10 Scale). Moderate-intensity aerobic exercise: jumps and static running at various speeds. RPE: 5–6 (Borg CR-10 Scale). Muscle strengthening exercises: 8–12 reps, 1 set, 8–10 exercises (hip flexion, extension, adduction, and abduction, knee flexion and extension, knee cycling, shoulder flexion, extension, and abduction, elbow flexion, extension, and pronation-supination, squatting). RPE: 5–6 (Borg CR-10 Scale). Static stretching exercises: stretching for 30–45s, 1 rep, 8–10 exercises (neck, trunk, and limbs). RPE: 3–4 (Borg CR-10 Scale). **Relaxation/cool-down phase:** Low-intensity exercise: slow walking, squatting and standing. RPE: 2–4 (Borg CR-10 Scale). Diaphragmatic breathing/ Floating. RPE: 0–2 (Borg CR-10 Scale). | **Warm-up phase:** Low-intensity aerobic exercise: walking forwards, sideways, and backwards. RPE: 3–4 (Borg CR-10 Scale). **Training phase:** Active range of motion exercises: spine, upper and lower limbs. RPE: 3–4 (Borg CR-10 Scale). Moderate-to-high intensity aerobic exercise: jumps and static running at various speeds. RPE: 6–8 (Borg CR-10 Scale). Muscle strengthening exercises: 8–12 reps, 2 sets, 8–10 exercises (hip flexion, extension, adduction, and abduction, knee flexion and extension, knee cycling, shoulder flexion, extension, and abduction, elbow flexion, extension, and pronation-supination, squatting). RPE: 6–7 (Borg CR-10 Scale). Static stretching exercises: stretching for 45–60s, 1 rep, 8–10 exercises (neck, trunk, and limbs). RPE: 3–4 (Borg CR-10 Scale). **Relaxation/cool-down phase:** Low-intensity exercise: slow walking, squatting and standing. RPE: 2–4 (Borg CR-10 Scale). Diaphragmatic breathing/ Floating. RPE: 0–2 (Borg CR-10 Scale). |

Notes: min, minutes; RPE, rate of perceived exertion; reps, repetitions; s, seconds.

Participants will be instructed to minimize tension in other body muscles and maintain their usual breathing pattern during stretching [64–67].

## Procedures

The experimental procedures will be conducted at the Municipal Pool of Paião, in Figueira da Foz. The research team will consist of a principal investigator, an examiner, a data analyst, and two intervention administrators. The SPIRIT schedule of the study protocol is presented in Fig 1.

**Feasibility study.** To assess the need for any adjustments in the experimental procedures and evaluate the feasibility of the clinical study methodology, a trial of the data collection protocol will be conducted with two to three individuals who meet the study's eligibility criteria.

**Table 2. Land-exercise program**

| Type | Frequency and duration | Intensity | |
|---|---|---|---|
| | | Week 1–4 | Week 5–8 |
| **Core stability exercises:** abdominal hollowing, side bridge (both sides), supine extension bridge, lower limb extension from prone (both sides), and alternate arm and leg raise from quadruped, performed in dry land. | 1 time per week. Session duration: 25–35 min (Core stability exercises: 15–20 min, and stretching exercises: 10–15 min). | **Isometric strength exercises:** contraction held for 7–8s, 8–10 reps, 1 set. Rest between reps: 3s Rest between exercises: 1min. RPE: 5–6 (Borg CR-10 Scale) | **Isometric strength exercises:** Contraction held for 9–10s, 10–12 reps, 1 set. Rest between reps: 3s Rest between exercises: 30s. RPE: 6–7 (Borg CR-10 Scale) |
| **Stretching exercises:** stretching the hamstrings, iliopsoas, piriformis, and tensor fasciae latae, performed in dry land. | | **Static stretching exercises:** stretching for 30–45s. 2 reps. Rest between reps: 5–10s. RPE: 5–6 (Borg CR-10 Scale) | **Static stretching exercises:** stretching for 45–60s. 3 reps. Rest between reps: 5–10s. RPE: 5–6 (Borg CR-10 Scale) |

Notes: min, minutes; s, seconds; reps, repetitions; RPE, rate of perceived exertion.

**Sample selection and characterization.** Participant recruitment has begun in September 2024 and is expected to be completed in January 2025. Participants will be invited and recruited to the study through posters, which will include a brief explanation of the study's aims and procedures. Interested individuals will complete a written selection and characterization questionnaire, and the principal investigator will provide detailed information about all procedures involved in the study.

**Randomization and Blinding.** Participants who meet eligibility criteria will be given an alphanumeric code to maintain data confidentiality and anonymity. Participants will be randomly assigned to either ALG or AG, with a 1:1 allocation. Randomization will be performed by the principal investigator using a block randomization method generated by a computer program to ensure equal sample sizes in both groups. The block size will not be disclosed to maintain concealment. Participants will be informed of their group allocation (ALG or AG) via telephone contact. The principal investigator will not be actively involved in the intervention, data collection or data analysis. Randomization will only be known to the physiotherapists responsible for the intervention and the principal investigator. The examiner will evaluate participants before the intervention and after 8 weeks of intervention, ensuring blinding to group allocation, and participants will be asked not to disclose this information to the examiner. Each intervention will have a physiotherapist responsible for its execution, who will be instructed not to reveal participants' group allocation to other research team members. Both data collector and data analyst will be unaware of the participants' group distribution and will also be considered blinded. Only the principal investigator will have full knowledge of the entire research flow.

**Data Collection.** After enrollment, participants will complete three evaluation phases: before the intervention (M0), after the intervention (M1), and 4 weeks after the end of the intervention (M2). Data collection started in November 2024 and is expected to be completed in April 2025. During data collection, participants will start each data collection session by filling out the questionnaires (Oswestry Disability Index, MOS Short Form Health Survey 36 Item, and Tampa Scale of Kinesiophobia-13) and the Visual Analog Scale for pain at rest, during movement, and at night, which will take approximately 30 minutes. Following this,

|  | STUDY PERIOD | | | | | |
|---|---|---|---|---|---|---|
|  | Baseline | | | Intervention | Follow-ups | |
|  | Visit 1 | Visit 2 | Visit 3 | Sessions | Visit 4 | Visit 5 |
| Week | -4 to -3 | -3 to - 1 | -1 to 0 | 0 to 8 | 9 | 12 |
| **Baseline Screening** |  |  |  |  |  |  |
| Informed consent, review inclusion and exclusion criteria | X |  |  |  |  |  |
| Sociodemographic information and PARQ | X |  |  |  |  |  |
| **Interventions** |  |  |  |  |  |  |
| Aquatic and land – exercise group |  |  |  | X |  |  |
| Aquatic – exercise group |  |  |  | X |  |  |
| **Assessments** |  |  |  |  |  |  |
| Visual Analogue Scale |  | X |  |  | X | X |
| Oswestry Disability Index |  | X |  |  | X | X |
| MOS Short Form Health Survey 36 |  | X |  |  | X | X |
| Tampa Scale of Kinesiophobia-13 |  | X |  |  | X | X |
| Modified-Modified Schober Test |  | X |  |  | X | X |
| **Randomization** |  |  | X |  |  |  |

**Fig 1. Schedule of enrolment, interventions and assessments.**

the physiotherapist responsible for the assessment will measure weight and height, and then assess lumbar spine mobility using the Modified-Modified Schober Test, taking the best of three measurements. To standardize data collection, each participant will be asked to stand barefoot on a paper where the outline of their feet will be drawn to ensure consistent positioning across all measurements. This procedure will be repeated three times, with a 30-second break between repetitions. Participants will be instructed to place their feet at hip-width apart and their upper limbs relaxed along the body in the test starting position [68,69]. Fig 2 shows the study design with key moments of the intervention and data collection.

## Outcome measures

**Sample Selection and Characterization Questionnaire.** A questionnaire will be developed to select and characterize the study sample. It will collect personal data (age and sex), anthropometric data (height and weight), symptom duration, comorbidities, and professional status.

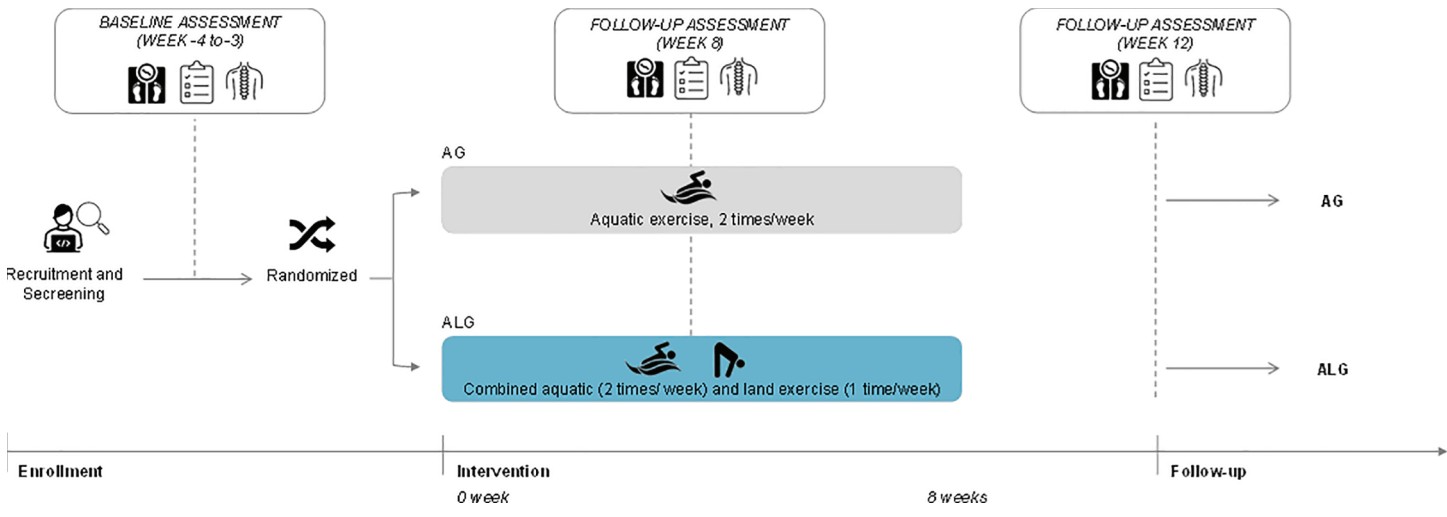

**Fig 2. Study design with key moments of the intervention and data collection.**

**Primary outcomes   Visual Analogue Scale (VAS).** The VAS is a unidimensional measure of pain intensity used to track its progression over time. This instrument consists of a 100-millimeter line, presented either vertically or horizontally, with an initial value of 0 indicating "No Pain" and an end value of 10 representing "Worst Pain." Participants are asked to mark a perpendicular line on the scale that corresponds to their level of pain. The scale is recorded on a single-use sheet of paper, and the value is obtained by measuring the distance from the origin (0) to the mark [70]. The VAS is a validated tool for assessing the intensity of CLBP, with moderate to strong construct validity (0.50–0.81), moderate to excellent test-retest reliability [Intraclass Correlation Coefficient (ICC) = 0.49–0.90], and a standard error of measurement (SEm) of 0.1% [71]. Additionally, the minimal clinically important difference (MCID) for this instrument is 20mm for CLBP [72].

**Oswestry Disability Index – version 2.0 (ODI).** The ODI is a tool used to measure and assess the disability caused by low back pain. It consists of 10 questions, each rated on a Likert scale from 0 to 5, where 0 represents no dysfunction and 5 represents the highest level of dysfunction. The final score is expressed as a percentage, indicating the level of disability experienced by the individual with low back pain: 0%–20% (minimal disability); 21%–40% (moderate disability); 41%–60% (severe disability); 61%–80% (very severe disability); 81%–100% (exaggerated symptoms) [73,74]. This instrument has been translated and validated for the Portuguese population, showing a good correlation with the MOS Short Form Health Survey 36 ($r = -0.59$ to $-0.75$) and excellent internal consistency (Cronbach's $\alpha = 0.95$) and reproducibility (ICC = 0.90) [75,76]. The MCID for this questionnaire was established at 10% [72,77].

**MOS Short Form Health Survey 36 Item v2 (SF-36 v2).** The SF-36 v2 questionnaire is an assessment tool primarily designed to measure and evaluate the health status and quality of life of populations and individuals, whether they have health conditions or not. It consists of 36 items, rated on a scale from 1 to 5 (Likert scale). The questionnaire is divided into 8 dimensions (physical functioning, physical role limitations, bodily pain, general health perceptions, energy/vitality, social functioning, emotional role limitations and mental health), which can be grouped into two components (physical health and mental health). Scores for each dimension are presented on a positively oriented scale from 0 (worst health state) to 100 (best health

state) [78]. The instrument was previously validated and translated for the Portuguese population, demonstrating good content validity ($r>0.401$) and moderate to excellent levels of internal consistency (Cronbach's $\alpha$ = 0.60–0.87) and test-retest reliability (ICC = 0.45-0.84) across its various domains [79,80]. For this instrument, the MCID has been established at 6 for mental health and 14 for physical health [81].

**Secondary outcomes   Tampa Scale of Kinesiophobia-13 (TSK-13).** The TSK-13 is a simple tool designed to quantify fear of movement or (re)injury. The questionnaire consists of 13 items, rated on a 4-point Likert scale (strongly disagree, disagree, agree, and strongly agree), with a final score ranging from 13 to 52 points [82]. The final classification of fear of movement can be categorized as subclinical (13–22), mild (23–32), moderate (33–42), and severe (43–52) [83]. The questionnaire has been translated for the Portuguese population, with construct validity established through its correlation with the VAS for pain intensity ($r = 0.691$) and confidence in movement ($r = -0.772$). The instrument shows excellent reliability
(ICC = 0.94–0.98) and internal consistency (Cronbach's $\alpha$ = 0.82) [84]. The MCID for this questionnaire has been established at 5.5 points [85].

**Modified-Modified Schober Test (MMST).** The MMST is a test originally designed to determine whether there is a reduction in lumbar spine range of motion due to various health conditions such as CLBP. It has subsequently been used to measure the effects of therapeutic interventions on lumbar mobility [86]. To perform this test, the participant should be in an upright position so that the examiner can mark both Posterior Superior Iliac Spines (PSIS) using a skin marker, then draw a horizontal line between them and mark a point at the intersection of this line with the spine (center of both marks). Next, the examiner should mark a second point 15 cm above the first. The participant is then instructed to bend the trunk as if trying to "touch their toes", and the examiner should measure the distance between the upper and lower lines again. An increase of less than 5 cm in the measured distance indicates a decrease in lumbar spine range of motion [87]. The MMST is a validated test for individuals with CLBP, showing a moderate correlation ($r = 0.67$) with the gold standard metric (X-ray) and excellent reliability, specifically in interclass correlations ($r = 0.91$) and intraclass correlations ($r = 0.95$) [88]. For this test, the minimal detectable difference is 1 cm [89].

**Anthropometry.** Body mass will be assessed using a portable scale (Inbody 270, USA) with an accuracy of 0.1kg, while participants will be asked to wear minimal clothes. Height will be measured using a portable stadiometer (SECA Bodymeter 208, Germany), with an accuracy of 1 millimeter, following standardized procedures [90]. Body Mass Index (BMI) will be calculated using the formula: BMI = (weight/height$^2$) [91]. Skeletal muscle mass (kg), fat mass (kg), bone mineral mass (%), and body fat (%) will be assessed using the tetrapolar bioimpedance (Inbody 270, USA). The impedance measurements will be performed according to the literature and manufactory recommendations [92].

## Adverse events

During the procedures, participants may experience some adverse events and should be aware of the potential risks during both the assessment phase and the intervention. To address this, a cardiovascular risk assessment will be conducted to contraindicate the performance of physical exercise/activity according to the American College of Sports Medicine flowchart, and the PAR-Q questionnaire will be administered. Participants classified as high risk will be excluded from the study to minimize the risk associated with physical exercise. For the remaining risk categories, the risk of a cardiovascular event is null or reduced [13]. Participants will be instructed to regularly check their skin and contact the principal investigator

if they experience a rash, redness, itching, or any other sign of an allergic reaction, although such adverse reactions are rare [37]. Thus, the physiotherapist will record any adverse events that may occur by asking participants if they have experienced any health issues (injuries, pain, etc.) since the previous session. All adverse events will be reviewed by the research team and appropriately referred for clinical treatment or rest.

## Participant adherence

Individual participation in each exercise session (both land and aquatic) will be recorded by the physiotherapist responsible for the intervention with a "0" if the participant does not attend or a "1" if the participant is present and participates. Strategies will be adopted to promote adherence to the exercises, including text messages and/or email [93]. If a participant misses two consecutive sessions, he/she will be contacted to resume the exercise sessions, as lack of adherence may affect the results obtained [94]. At the end of the interventions, participants who do not achieve at least 80% of the total sessions will be excluded from data analysis [91,95].

## Statistical analysis

An inspection of the data will be made for missing values or univariate outliers. A priori power analysis through G$^*$Power (3.1.9.7) [96] and ANOVA repeated measures, within-between interaction will be use to determine the required sample size considering the following input parameters: (effect size $f = 0.4$; $\alpha$ err prob = 0.05; statistical power = 0.95) [97]. Data analyses will be performed using SPSS Statistics version 29.0 (SPSS Inc., IBM Company, Chicago, Illinois, USA). Descriptive statistics (mean $\pm$ SD) will be performed for all variables in the analysis. Assuming data normality, a $2 \times 3$ within-between interaction factorial ANOVA will be applied for intra and inter group comparisons. To control for Type I errors, multiple comparisons will be adjusted using the Bonferroni correction (i.e., alpha level divided by the number of tests) as suggested by several authors (e.g., Ho, 2014) [98].The magnitude of the global effect size will be calculated using Cohen's effect size. The Cohen's $d$ effect size is categorized according to the following criteria: $d = < 0.20$ (small), 0.21–0.79 (medium) and > 0.80 (large) [99–101]. Significance level set at $\rho \leq 0.05$. Full statistical analysis and results are expected to be completed in May 2025.

## Discussion

This study will investigate and compare the effectiveness of a water-based and land-based exercise program versus a solely water-based program on pain levels, functional disability, quality of life, lumbar mobility, and fear of movement in adults with CLBP. Additionally, an effort will be made to assess the effect of the cessation of exercise on the mentioned variables, 4 weeks after the end of the intervention. A minimum of 15 participants per group is expected to ensure the calculated sample power. However, a higher number of participants should be ensured, as it is known that approximately 50% of participants drop out within the first six months due to personal characteristics, environmental factors, or issues with the programs [55]. On the other hand, in order to maintain the methodological quality of the study, according to the Physiotherapy Evidence Database (PEDro) Scale, the variables should be measured at the end in more than 85% of the participants who were initially randomized [102].

Previously, an author conducted a RCT with 12 participants (6 in the experimental group and 6 in the control group) in which the effect of an aquatic exercise program (experimental)

was compared with that of a land-based program (control) on the level of pain and mobility of the lumbar spine [41]. At the end of the interventions (6 weeks), no statistically significant differences were observed in pain (VAS) ($\rho$=0.523), nor in the lumbar spine mobility test (Modified-Modified Schober Test) ($\rho$=0.56). Another author also studied in a RCT the differences between a group of participants who engaged in an aquatic exercise program (experimental) and a group who engaged in a land-based program (control) with 65 individuals with LBP (32 in the experimental group and 33 in the control group) [42]. After 4 weeks of intervention, statistically significant differences were observed favoring the experimental intervention in the ODI score ($\rho < 0.001$) and in the physical function ($\rho < 0.001$) and physical performance ($\rho < 0.001$) domains of the SF-36 questionnaire. No statistically significant differences were found between the groups in rest pain, pain during movement, or nocturnal pain (VAS), in lumbar mobility test (Modified-Modified Schober Test), or in the remaining domains of the SF-36 questionnaire.

The study (RCT) with the most similar methodology was conducted with 66 participants (33 in each group) and aimed to examine the effect of an experimental aquatic and land-based exercise program compared to a control program of exclusively land-based (home-based) [43]. After two weeks of intervention, statistically significant differences were observed favoring the experimental intervention in pain: VAS at rest ($\rho < 0.001$), during exercise ($\rho < 0.001$), and overall in the lumbar spine ($\rho < 0.001$); in the ODI score ($\rho = 0.045$); and in the SF-36 domains of physical performance ($\rho = 0.033$), mental health ($\rho = 0.021$), pain ($\rho = 0.014$), and general health ($\rho = 0.05$). No statistically significant differences were found between the groups in the remaining domains of the SF-36 questionnaire or in the Modified-Modified Schober Test.

No study with a similar methodology has reported results for the TSK-13 questionnaire to date. Thus, it is challenging to predict the results of the study, as the methodologies differ and the sample sizes and intervention durations vary considerably among authors. It is not possible to project whether the group that will undertake the additional land-based exercise session will achieve better outcomes than the experimental group at the end of the intervention, in the studied variables. However, the variables most sensitive to the presented interventions appear to be pain, measured by VAS, functional disability measured by ODI, and some domains of the SF-36 related to pain and physical performance. These are likely to be the variables with the greatest variability at the end of the study. On the other hand, it is difficult to predict the trend of changes in the studied variables after 1 month of exercise cessation due to the lack of reference literature.

Thus, this study aims to provide implications for the practice and prescription of physical exercise and effective strategies that may contribute to potential benefits for individuals with LBP. Like all studies, this one will have some limitations: it will be impossible to fully control activities outside of the protocol, which may affect the studied variables either positively or negatively, and aquatic exercise limits the calculation of exercise intensity, which will be monitored only through perceived exertion.

## Dissemination of results

Upon completion of the study, regardless of the direction and magnitude of the results, they will be published. The results will be communicated to the participants and their families, the scientific community, and the general public, through oral presentations and publications in recognized journals in the field.

Authorship of the manuscripts resulting from this study will be based on: significant contributions to the study conceptualization or design, formal analysis and/or interpretations of the data, and revising and editing the manuscript critically.

## Supporting information

**S1 File. Filled SPIRIT checklist.**
(DOC).

## Acknowledgments

The authors would like to thank the organizations that collaborated in the development of the proposed study: ESECS-Polytechnic of Leiria, Department of Sports and Well-Being of Polytechnic Institute of Castelo Branco and Municipal Pool of Paião.

## Author contributions

**Conceptualization:** Joana Borges, Tiago Pastilha.

**Formal analysis:** Diogo Monteiro, Fernanda M. Silva.

**Funding acquisition:** Diogo Monteiro.

**Investigation:** Joana Borges, Tiago Pastilha.

**Methodology:** Joana Borges, Diogo Monteiro, Fernanda M. Silva, Tiago Pastilha, Pedro Duarte-Mendes.

**Project administration:** Joana Borges, Tiago Pastilha.

**Resources:** Pedro Duarte-Mendes.

**Supervision:** Diogo Monteiro, Fernanda M. Silva, Miguel Jacinto, Pedro Duarte-Mendes.

**Validation:** Diogo Monteiro, Pedro Duarte-Mendes.

**Writing – original draft:** Joana Borges, Tiago Pastilha.

**Writing – review & editing:** Diogo Monteiro, Fernanda M. Silva, Miguel Jacinto, Pedro Duarte-Mendes.

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
