## [Decision Letter · Decision Letter 0]

2 Feb 2025

PONE-D-25-01668Effects of a land and aquatic exercise-based program on pain, mobility and quality of life in patients with chronic low back pain: A study protocol for a randomized controlled trialPLOS ONE

Dear Dr. Borges,

Thank you for submitting your manuscript to PLOS ONE. After careful consideration, we feel that it has merit but does not fully meet PLOS ONE’s publication criteria as it currently stands. Therefore, we invite you to submit a revised version of the manuscript that addresses the points raised during the review process.

We look forward to receiving your revised manuscript.

Kind regards,

Shabnam ShahAli, Ph.D.

Academic Editor

PLOS ONE

Journal Requirements:

2. We note that you have selected “Clinical Trial” as your article type. PLOS ONE requires that all clinical trials are registered in an appropriate registry (the WHO list of approved registries is at      https://www.who.int/clinical-trials-registry-platform/network/primary-registries" https://www.who.int/clinical-trials-registry-platform/network/primary-registries and more information on trial registration is at http://www.icmje.org/about-icmje/faqs/clinical-trials-registration/).

Please state the name of the registry and the registration number (e.g. ISRCTN or ClinicalTrials.gov) in the submission data and on the title page of your manuscript.

a) Please provide the complete date range for participant recruitment and follow-up in the methods section of your manuscript.

b) If you have not yet registered your trial in an appropriate registry, we now require you to do so and will need confirmation of the trial registry number before we can pass your paper to the next stage of review. Please include in the Methods section of your paper your reasons for not registering this study before enrolment of participants started. Please confirm that all related trials are registered by stating: “The authors confirm that all ongoing and related trials for this drug/intervention are registered”.

Please see http://journals.plos.org/plosone/s/submission-guidelines#loc-clinical-trials for our policies on clinical trials.

“This work was funded by National Funds by FCT - Foundation for Science and 418

Technology under the following project UIDB/04045/2020 419

(https://doi.org/10.54499/UIDB/04045/2020).”

“This work was funded by National Funds by FCT - Foundation for Science and Technology under the following project UIDB/04045/2020 (https://doi.org/10.54499/UIDB/04045/2020). The authors would like to thank the organizations that collaborated in the development of the proposed study: ESECS-Polytechnic of Leiria, Department of Sports and Well-Being of Polytechnic Institute of Castelo Branco and Municipal Pool of Pai˜ao”.

“This work was funded by National Funds by FCT - Foundation for Science and 418

Technology under the following project UIDB/04045/2020 419

(https://doi.org/10.54499/UIDB/04045/2020).”

6. In the online submission form, you indicated that your data will be submitted to a repository upon acceptance.  We strongly recommend all authors deposit their data before acceptance, as the process can be lengthy and hold up publication timelines. Please note that, though access restrictions are acceptable now, your entire minimal  dataset will need to be made freely accessible if your manuscript is accepted for publication. This policy applies to all data except where public deposition would breach compliance with the protocol approved by your research ethics board. If you are unable to adhere to our open data policy, please kindly revise your statement to explain your reasoning and we will seek the editor's input on an exemption.

7. Your ethics statement should only appear in the Methods section of your manuscript. If your ethics statement is written in any section besides the Methods, please move it to the Methods section and delete it from any other section. Please ensure that your ethics statement is included in your manuscript, as the ethics statement entered into the online submission form will not be published alongside your manuscript.

8. We note that the original protocol file you uploaded contains a confidentiality notice indicating that the protocol may not be shared publicly or be published. Please note, however, that the PLOS Editorial Policy requires that the original protocol be published alongside your manuscript in the event of acceptance. Please note that should your paper be accepted, all content including the protocol will be published under the Creative Commons Attribution (CC BY) 4.0 license, which means that it will be freely available online, and any third party is permitted to access, download, copy, distribute, and use these materials in any way, even commercially, with proper attribution.

Reviewers' comments:

Reviewer's Responses to Questions

**Comments to the Author**

1. Does the manuscript provide a valid rationale for the proposed study, with clearly identified and justified research questions?

Reviewer #1: Yes

Reviewer #2: Yes

Reviewer #3: Yes

2. Is the protocol technically sound and planned in a manner that will lead to a meaningful outcome and allow testing the stated hypotheses?

Reviewer #1: Yes

Reviewer #2: Partly

Reviewer #3: Yes

3. Is the methodology feasible and described in sufficient detail to allow the work to be replicable?

Reviewer #1: Yes

Reviewer #2: Yes

Reviewer #3: Yes

4. Have the authors described where all data underlying the findings will be made available when the study is complete?

Reviewer #1: Yes

Reviewer #2: Yes

Reviewer #3: Yes

5. Is the manuscript presented in an intelligible fashion and written in standard English?

Reviewer #1: Yes

Reviewer #2: Yes

Reviewer #3: Yes

6. Review Comments to the Author

You may also provide optional suggestions and comments to authors that they might find helpful in planning their study.

Reviewer #1: Good protocol paper with enough information. However there are some points needs to modify.

firstly, I did not satisfy with your work novelty. Are the outcome measures the only things that show your paper novelty?

Second, did you upload SPIRIT checklist to editor board with each item completed?

Reviewer #2: I appreciate the opportunity to review this study. The authors have devoted significant time and effort to this study. I have provided some comments that should be addressed to improve the quality of the manuscript.

Introduction

1- Please add your specific hypotheses at the end of introduction.

Methods

2- The pre-specified outcome measures were not defined in terms of primary or secondary. Please define primary and secondary outcome measures in the manuscript.

3- The sample size should be calculated based on the pre-specified primary outcome. Did the authors consider the pain intensity (VAS) as the primary outcome in this study?

Reviewer #3: The manuscript has been accepted for publication following a comprehensive review. The reviewers found that it adheres to the journal's standards, with commendable methodology and data analysis.

Additionally, all required revisions have been properly addressed, enhancing the quality and clarity of the work.

7. PLOS authors have the option to publish the peer review history of their article (what does this mean?). If published, this will include your full peer review and any attached files.

Reviewer #1: No

Reviewer #2: No

Reviewer #3: **Yes: **Atieh Nazem

---

## [Author Response · Author response to Decision Letter 1]

5 Feb 2025

Dear Professor Shabnam ShahAli,

My colleagues and I would like to express our gratitude for the opportunity to resubmit our manuscript, Effects of a land and aquatic exercise-based program on pain, mobility and quality of life in patients with chronic low back pain: A study protocol for a randomized controlled trial" by Joana Borges, Diogo Monteiro, Fernanda M. Silva, Miguel Jacinto, Tiago Pastilha and Pedro Duarte-Mendes to Plos One. We greatly appreciate the helpful comments from the academic editor and reviewers. We have made every effort to incorporate their suggestions and address all their feedback. We believe these revisions have significantly improved the manuscript.

The manuscript was previously submitted to Plos One (PONE-D-25-01668) and changes were made in order to clarify the issues presented by the Academic Editor.

If you need any further information, please feel free to contact us.

Best Regards,

Joana Borges and co-authors

---

## [Decision Letter · Decision Letter 1]

13 Feb 2025

PONE-D-25-01668R1Effects of a land and aquatic exercise-based program on pain, mobility and quality of life in patients with chronic low back pain: A study protocol for a randomized controlled trialPLOS ONE

Dear Dr. Borges,

Thank you for submitting your manuscript to PLOS ONE. After careful consideration, we feel that it has merit but does not fully meet PLOS ONE’s publication criteria as it currently stands. Therefore, we invite you to submit a revised version of the manuscript that addresses the points raised during the review process.

We look forward to receiving your revised manuscript.

Kind regards,

Shabnam ShahAli, Ph.D.

Academic Editor

PLOS ONE

Journal Requirements:

Reviewers' comments:

Reviewer's Responses to Questions

**Comments to the Author**

1. Does the manuscript provide a valid rationale for the proposed study, with clearly identified and justified research questions?

Reviewer #1: Yes

Reviewer #2: Yes

2. Is the protocol technically sound and planned in a manner that will lead to a meaningful outcome and allow testing the stated hypotheses?

Reviewer #1: Yes

Reviewer #2: Yes

3. Is the methodology feasible and described in sufficient detail to allow the work to be replicable?

Reviewer #1: Yes

Reviewer #2: Yes

4. Have the authors described where all data underlying the findings will be made available when the study is complete?

Reviewer #1: Yes

Reviewer #2: Yes

5. Is the manuscript presented in an intelligible fashion and written in standard English?

Reviewer #1: Yes

Reviewer #2: Yes

6. Review Comments to the Author

You may also provide optional suggestions and comments to authors that they might find helpful in planning their study.

Reviewer #1: Thanks for your good paper with enough information. The changes are acceptable. However, it is better to check out each item in spirit checklist. And it may make your paper more valuable if you add more outcome measures.

Reviewer #2: Dear authors,

Well done. My recommendations were addressed. My decision is now ACCEPTANCE.

Good Luck

7. PLOS authors have the option to publish the peer review history of their article (what does this mean?). If published, this will include your full peer review and any attached files.

Reviewer #1: **Yes: **Laleh Abadi marand

Reviewer #2: No

---

## [Author Response · Author response to Decision Letter 2]

14 Feb 2025

Dear Academic Editor Shabnam ShahAli,

Dear Reviewers,

We are grateful for the opportunity to resubmit our manuscript to PLOS ONE. The feedback from reviewers was helpful, and we have responded to their comments.

Attached below are the reviewers' comments along with a summary of the changes we made. Additionally, we have submitted an updated version of the manuscript with all revisions highlighted in yellow. Please feel free to reach out if you need any further information.

We confirm that the reference list has been thoroughly reviewed and is both complete and accurate.

Review Comments to the Author

Reviewer #1: Thanks for your good paper with enough information. The changes are acceptable. However, it is better to check out each item in spirit checklist. And it may make your paper more valuable if you add more outcome measures.

Authors' response: Thank you very much for your comment and contribution to our paper in order to enhance our work and make it more valuable. Regarding the suggestion to add more outcome measures, we have carefully considered this possibility. However, we believe that the outcomes already included are appropriate and sufficient to address the study objectives and provide relevant results.

The SPIRIT checklist was updated and we added information about the point 5d. We added the section “Author Contributions” to our manuscript in page 13, lines 432-446.

Reviewer #2: Dear authors,

Well done. My recommendations were addressed. My decision is now ACCEPTANCE.

Good Luck

Authors' response: Thank you very much for your comment and contributions.

Joana Borges and Co-authors

---

## [Editor Report · Decision Letter 2]

20 Feb 2025

PONE-D-25-01668R2Effects of a land and aquatic exercise-based program on pain, mobility and quality of life in patients with chronic low back pain: A study protocol for a randomized controlled trialPLOS ONE

Dear Dr. Borges,

Thank you for submitting your manuscript to PLOS ONE. After careful consideration, we feel that it has merit but does not fully meet PLOS ONE’s publication criteria as it currently stands. Therefore, we invite you to submit a revised version of the manuscript that addresses the points raised during the review process.

We look forward to receiving your revised manuscript.

Kind regards,

Shabnam ShahAli, Ph.D.

Academic Editor

PLOS ONE

Journal Requirements:

Additional Editor Comments:

The paper is generally well written and the concerns of the previous reviewers have been addressed and appear reasonable.

However, there are some concerns remaining from the statistical design and analysis perspective.

1. Please check other references to be sure that the effect size of 1.079 for the VAS being used for this study’s sample size consideration is realistic. It does appear rather large and based on only one reference.

2. The inflation of 15% for the sample looks rather arbitrary. What is this based on? The sample size is small enough with the assumed large effect size and one does not want to come up short after such an effort.

3. There are three primary endpoints with an overall 0.05 significance level being used for the study. This involves multiple comparisons and thus needs an adjustment to the p-value or the type one error is inflated with multiple endpoints being considered. One may consider a Bonferroni type or other type of correction such as Holm-Bonferroni to the alpha level being used for each test. Likewise, there are three secondary endpoints and one could challenge these results if a level of 0.05 is maintained. Ultimately, this could affect the actual sample size required for the entire study.

---

## [Author Response · Author response to Decision Letter 3]

23 Feb 2025

Dear Editor,

My colleagues and I would like to thank you for the opportunity to resubmit our manuscript to PLOS ONE. We found that the academic reviewers’ and editor comments were very helpful, and we have done our best to incorporate all their suggestions and reply to the comments. We believe that this has made a significant contribution to the overall quality of the manuscript.

Our responses are attached at the bottom of this letter. We have also submitted an updated version of our manuscript with all the changes highlighted in yellow. If you require any additional information, please do not hesitate to get in touch with us.

Thank you for considering our manuscript.

On behalf of all authors, yours sincerely,

Joana Borges

---

## [Editor Report · Decision Letter 3]

27 Feb 2025

Effects of a land and aquatic exercise-based program on pain, mobility and quality of life in patients with chronic low back pain: A study protocol for a randomized controlled trial

PONE-D-25-01668R3

Dear Dr. Borges,

We’re pleased to inform you that your manuscript has been judged scientifically suitable for publication and will be formally accepted for publication once it meets all outstanding technical requirements.

Kind regards,

Shabnam ShahAli, Ph.D.

Academic Editor

PLOS ONE

---

## [Editor Report · Acceptance letter]

PONE-D-25-01668R3

PLOS ONE

Dear Dr. Borges,

I'm pleased to inform you that your manuscript has been deemed suitable for publication in PLOS ONE. Congratulations! Your manuscript is now being handed over to our production team.

Kind regards,

on behalf of

Dr. Shabnam ShahAli

Academic Editor

PLOS ONE